# Integration of QTL Mapping and Whole Genome Sequencing Identifies Candidate Genes for Alkalinity Tolerance in Rice (*Oryza sativa*)

**DOI:** 10.3390/ijms231911791

**Published:** 2022-10-04

**Authors:** Lovepreet Singh, Sapphire Coronejo, Rajat Pruthi, Sandeep Chapagain, Prasanta K. Subudhi

**Affiliations:** School of Plant, Environmental, and Soil Sciences, Louisiana State University Agricultural Center, Baton Rouge, LA 70803, USA

**Keywords:** abiotic stress, genotyping-by-sequencing, Na^+^/K^+^ ratio, *Oryza sativa*, quantitative trait loci, seedling stage

## Abstract

Soil alkalinity is an important stressor that impairs crop growth and development, resulting in reduced crop productivity. Unlike salinity stress, research efforts to understand the mechanism of plant adaptation to alkaline stress is limited in rice, a major staple food for the world population. We evaluated a population of 193 recombinant inbred lines (RIL) developed from a cross between Cocodrie and N22 under alkaline stress at the seedling stage. Using a linkage map consisting of 4849 SNP markers, 42 additive QTLs were identified. There were seven genomic regions where two or more QTLs for multiple traits colocalized. Three important QTL clusters were targeted, and several candidate genes were identified based on high impact variants using whole genome sequences (WGS) of both parents and differential expression in response to alkalinity stress. These genes included two expressed protein genes, the glucan endo-1,3-beta-glucosidase precursor, F-box domain-containing proteins, double-stranded RNA-binding motif-containing protein, aquaporin protein, receptor kinase-like protein, semialdehyde hydrogenase, and NAD-binding domain-containing protein genes. Tolerance to alkaline stress in Cocodrie was most likely due to the low Na^+^/K^+^ ratio resulting from reduced accumulation of Na^+^ ions and higher accumulation of K^+^ in roots and shoots. Our study demonstrated the utility of integrating QTL mapping with WGS to identify the candidate genes in the QTL regions. The QTLs and candidate genes originating from the tolerant parent Cocodrie should be targeted for introgression to improve alkalinity tolerance in rice and to elucidate the molecular basis of alkali tolerance.

## 1. Introduction

Rice is a staple food for half of the world population. As the human population is expected to reach 9.7 billion by 2050, there is an urgent need to increase rice production by 50% to ensure global food security [1]. On the other hand, various abiotic stresses, such as drought, alkalinity, salinity, and temperature extremes, adversely affect crop production, resulting in significant yield losses globally [2]. It is estimated that half of the 830 million hectares of salt-affected land are alkaline globally [3] and soil salinization will impact 50% of all agricultural land by 2050 [4]. Salinealkaline stress can be characterized into two categories. Saline stress occurs due to high concentrations of neutral salts, NaCl and Na_2_SO_4_ [5], whereas alkaline stress is caused by an excess of carbonated salts (Na_2_CO_3_, NaHCO_3_) and high pH ranging from 8.5 to 11 [6]. Therefore, plants under alkaline stress not only suffer from osmotic stress and ionic imbalances, but also from the high pH. The problem of soil alkalization has been increasing due to climate change, improper fertilization, and use of poor-quality irrigation water.

Rice is sensitive to saline–alkaline stress. Osmotic stress and ionic imbalances damage the root cells, disrupt physiological mechanisms, inhibit plant growth, and significantly reduce rice yield [7]. High pH around the rhizosphere disturbs ionic balance and decreases nutrient availability due to the precipitation of essential nutrients such as iron and phosphorus, leading to reduced growth and development in rice plants [8]. Increase in absorption of Na^+^ by plants under alkaline stress disrupts the homeostasis of other minerals such as K^+^ and alters cytoplasmic ionic strength, which subsequently results in the disruption of cellular metabolism [9]. The tolerance mechanisms for salinity in rice have been extensively studied [10,11,12], including map-based cloning of genes *SKC1* and *DST* [13,14]. In contrast, research on plant response and adaptation to an alkaline environment has been limited, even though alkaline stress is more damaging to crops than saline stress [15]. Therefore, understanding alkalinity tolerance mechanisms in rice is crucial to improve its productivity.

Alkaline stress tolerance involves three mechanisms—osmotic stress tolerance, Na^+^ exclusion, and high pH tolerance. The root is the first plant organ that responds to saline–alkali stress, and therefore, plays a critical role in tolerance. The anatomy of the root system under osmotic stress determines plant performance and its capacity to acquire water. Osmotic stress or physiological drought inhibits the water absorption capacity of roots due to the high Na^+^ concentration around the rhizosphere [16]. The proliferated root system helps uptake of water and nutrients under a stress environment and improves crop yield [17]. In alkaline soils, availability of Fe to plants is limited due to the insoluble hydroxide and oxide form of soil under high pH conditions [18]. Plants with efficient Fe acquisition capacities under alkaline stress exhibit tolerance to high pH [19]. Na^+^ exclusion is a critical factor in conferring tolerance to alkaline stress because it reduces the amount of Na^+^ in the cytosol of cells and prevents its accumulation within leaf blades, leading to effective maintenance of the Na^+^/K^+^ ratio [20]. The Na^+^ exclusion mechanism and the role of Fe and other ion transporters under alkaline stress are not fully understood.

Quantitative trait loci (QTL) mapping is widely used for identifying genomic regions responsible for complex quantitative traits. The complexity of salinity–alkalinity tolerance was evident from several studies that identified multiple QTLs controlling morpho-physiological traits associated with alkalinity tolerance in rice [21,22,23,24,25,26]. A major QTL, *qAT11*, for alkali tolerance in *japonica* rice was identified on chromosome 11 [15], and haplotype analysis and gene expression analysis narrowed this region to three genes (LOC_Os11g37300, LOC_Os11g37320, and LOC_Os11g37390). In a genome-wide association study, a major QTL for alkali tolerance and shoot sodium and potassium concentrations with a phenotypic variation of ~14% colocalized on chromosome 3, where the gene OsIRO3 was identified [25]. Similarly, a major QTL for shoot Na^+^ concentration with 21% of phenotypic variation was detected [24]. Liang et al. [23] mapped seven additive QTLs for dead leaf rate (DLR) and dead seedling rate (DSR) during the seedling stage under salinity or alkalinity stress in a recombinant inbred line (RIL) population. In a backcross inbred line (BIL) population from an *indica* x *japonica* cross, different sets of QTLs were identified under control, saline, and alkaline stress conditions [26].

Large-effect QTLs with consistent effects in multiple environments are prerequisite to introgress alkalinity tolerance attributes in breeding programs. Next generation sequencing (NGS) can be exploited to dissect the QTL regions for discovering desirable alleles for alkalinity tolerance. Specifically, the analysis using whole genome sequencing (WGS) may provide beneficial allelic variants from contrasting parents [27]. Integration of QTL mapping and NGS technology can provide accurate and useful information on the candidate genes responsible for tolerance in the QTL regions.

In this study, we used a RIL population developed from a cross between Cocodrie and N22 to identify candidate genes for alkalinity tolerance in rice. Genotype-by-sequencing (GBS) was used to construct a high-resolution genetic map for the identification of QTLs, and WGS helped find the allelic variants in the QTL genomic regions. Candidate genes identified in this study could serve as targets for future breeding programs to improve alkalinity tolerance in rice.

## 2. Results

### 2.1. Phenotypic Evaluation

Phenotypic evaluation of alkalinity tolerance at the seedling stage exhibited wide variation in all morphological and physiological traits (Table 1; Figure 1). There were significant differences between Cocodrie and N22, as well as among the RILs, for all traits under study. The RIL means for all traits, except shoot length (SHL) and root-to-shoot ratio (RSR), were in-between the parental means under alkalinity stress (Table 1, Figure 2). Mean values of alkalinity tolerance score (AKT), shoot Na^+^ concentration (SNC), root Na^+^ concentration (RNC), shoot Na^+^ to K^+^ ratio (SNK), and root Na^+^ to K^+^ ratio (RNK) were lower in Cocodrie compared with N22. On the other hand, N22 exhibited lower values of chlorophyll content (CHL), root length (RTL), SHL, shoot K^+^ concentration (SKC), and root K^+^ concentration (RKC). The mean AKT scores were 4.3, 9.0, and 5.0 for Cocodrie, N22, and RILs, respectively. The SNC and SKC means of Cocodrie were closer to the RIL mean than N22 mean. All traits were normally distributed in the RIL population (Figure 2) and transgressive segregants were observed on both sides of the distribution. Low heritability was observed for CHL, whereas AKT and RTL showed medium heritability, and the rest of the traits showed high heritability (Table 1).

There was no significant difference between Cocodrie and N22 for any traits except RNC under the control environment (Table 1). Both parents showed a reduction in CHL, SHL, and RTL under stress environment. N22 showed a higher reduction in CHL, SHL, and RTL compared with Cocodrie (Figure 3). Increased RSR was observed under alkalinity stress in Cocodrie, but N22 showed significant decrease in RSR. Although both parents showed reduction in SKC and RKC under alkalinity stress, N22 experienced higher reductions in SKC than Cocodrie. Although there was increase in SNC, RNC, SNK, and RNK in both parents under alkaline stress, the increases were relatively higher in N22 compared with Cocodrie.

### 2.2. Correlations among Traits

There were significant correlations among most of the traits under alkaline stress (Table 2). Alkalinity tolerance score was negatively correlated to CHL, RTL, SKC, and RKC, but was positively correlated to SNC, RNC, SNK, and RNK. Chlorophyll content was positively correlated with SKC, RKC, and RSR, whereas it was negatively correlated with AKT, SNC, RNC, SNK, and RNK. Shoot length was positively correlated with SKC and negatively correlated with RSR, RNC, and RKC. Root length showed negative correlations with SNK and RNK and positive correlations with RSR, SKC, and RKC.

### 2.3. QTL Analysis

Inclusive composite interval mapping (ICIM) identified 42 additive QTLs under alkaline stress (Table 3, Figure 4) and 26 additive QTLs under non-stress environment (Appendix A). However, simple interval mapping (IM) detected 30 additive QTLs for alkaline stress (Appendix A) and 29 additive QTLs for non-stress conditions (Appendix A).

#### 2.3.1. Alkalinity Tolerance Score (AKT)

Inclusive composite interval mapping detected five QTLs (*qAKT3.03*, *qAKT5.008*, *qAKT8.002*, *qAKT9.19*, and *qAKT10.18*) (Table 3). Among these QTLs, *qAKT9.19* was a major effect QTL with a LOD score of 5.9 and 12% contribution toward phenotypic variation (PVE). The remaining QTLs were small-effect QTLs with 4–6% of total PVE. The N22 allele was responsible for increasing the mean AKT score in the case of four QTLs (*qAKT3.03*, *qAKT8.002*, *qAKT9.19*, and *qAKT10.18*), whereas it was the Cocodrie allele at *qAKT5.008*. In interval mapping, one major QTL, *qAKT9.20*, with 10% of total phenotypic variation was identified (Appendix A). This QTL was likely the same as *qAKT9.19* identified in ICIM.

#### 2.3.2. Chlorophyll Content (CHL)

There were three and four additive QTLs identified for chlorophyll content under alkaline stress by ICIM and IM, respectively (Table 3 and Appendix A). The *qCHL9.20* was a major effect QTL, detected by both ICIM and IM with a high LOD score and contribution of 18% toward PVE. The *qCHL1.37*, identified by both ICIM and IM, accounted for 6% of total PVE. In both cases, the allele for increasing the trait mean was contributed by N22. Another small-effect additive QTL, *qCHL8.002*, explained 5% of total PVE. No QTL was detected for chlorophyll content under control conditions.

#### 2.3.3. Shoot Length (SHL), Root Length (RTL), and Root-to-Shoot Ratio (RSR)

There were seven and six additive QTLs for shoot length under alkaline stress detected by ICIM and IM, respectively (Table 3 and Appendix A). The *qSHL1.38*, detected by both ICIM and IM, accounted for phenotypic variation of 44 and 23%, respectively. Another large-effect QTL, *qSHL1.37*, detected by IM, was responsible for 16% of total phenotypic variation. The N22 allele was responsible for increasing shoot length for the large-effect and small-effect additive QTLs, *qSHL3.13*, *qSHL7.05*, and *qSHL7.28*, whereas the Cocodrie allele contributed toward an increased mean for *qSHL1.03*, *qSHL6.26*, and *qSHL8.27*. Five and six additive QTLs were identified for shoot length under the non-stress environment by ICIM and IM, respectively (Appendix A). The N22 allele increased mean SHL in the case of eight QTLs and the Cocodrie allele in the rest.

Two additive QTLs (*qRTL3.28* and *qRTL7.26*) with 8% phenotypic variation were identified under alkaline stress by ICIM (Table 3). The *qRTL3.28* was also detected by IM under alkaline stress (Appendix A). The N22 allele at the *qRTL7.26* and *qRTL3.28* was responsible for increased root length, whereas Cocodrie allele was desirable in the case of *qRTL3.28*. Two additive QTLs explaining about 4–8% of total phenotypic variation were common in both IM and ICIM analyses under the non-stress environment (Appendix A).

There were four and three additive RSR QTLs on chromosomes 1, 3, and 8 detected by ICIM and IM, respectively (Table 3 and Appendix A). The *qRSR1.35* accounted for 21% of total PVE. The allele for increased RSR for this QTL was from Cocodrie. A large-effect QTL, *qRSR1.37*, detected by IM, explained 12% phenotypic variation and had the desirable allele from Cocodrie. Other additive QTLs were small-effect QTLs. Both N22 and Cocodrie alleles of these QTLs contributed toward increased means. Under the non-stress environment, two additive QTL were identified on chromosomes 1 and 3 (Appendix A).

#### 2.3.4. Shoot Na^+^ Concentration (SNC) and Root Na^+^ Concentration (RNC)

Four additive QTLs (*qSNC4.16*, *qSNC8.002*, *qSNC9.19*, and *qSNC12.19*) were identified by ICIM under alkaline stress (Table 3), and three of these QTLs were the same as the additive QTLs detected by IM (Appendix A). The large-effect QTL, *qSNC9.19*, was detected with a LOD score of 4.8 and total PVE of 12%. Other QTLs were small-effect QTLs with 4–8% of total phenotypic variation. N22 alleles at these QTLs contributed to increased SNC. Five additive SNC QTLs were identified by ICIM and IM under non-stress conditions (Appendix A).

Three additive QTLs were identified under alkaline stress by ICIM (Table 3). The *qRNC9.19* accounted for 11% of total phenotypic variation, whereas *qRNC12.19* and *qRNC8.002* contributed only 5 and 9% of the phenotypic variation, respectively. The N22 allele was responsible for increased means in the cases of *qRNC8.002* and *qRNC9.19*, whereas it was Cocodrie allele in the case of *qRNC12.19.* All three QTLs were also detected by IM (Appendix A). None of the QTLs were detected under the non-stress environment for RNC.

#### 2.3.5. Shoot K^+^ Concentration (SKC) and Root K^+^ Concentration (RKC)

There were three additive QTLs for SKC under alkaline stress detected by ICIM (Table 3) and only one small-effect QTL, *qSKC8.002*, accounting for 8% of phenotypic variation was common with IM analysis (Appendix A). The *qSKC10.18* explained 13% of total phenotypic variation. Cocodrie alleles were responsible for increased means for all the additive QTLs. Under non-stress conditions, five and six additive QTLs were detected for SKC by ICIM and IM (Appendix A). One major QTL, *qSKC1.38*, was detected with a LOD score of 21.7 and PVE of 38%. The N22 allele of this QTL increased the mean effect of this trait.

Four and two QTLs were detected for RKC under alkaline stress by ICIM and IM, respectively (Table 3 and Appendix A). Two of these QTLs (*qRKC8.002* and *qRKC9.19*) were also identified by IM. All these were small-effect QTLs with contributions of 6–8% toward PVE, with the exception of *qRKC9.19*, which explained 10% of PVE. The Cocodrie alleles at all the QTLs were responsible for increasing root K^+^ concentration. A total of four and five additive QTLs were identified under non-stress conditions (Appendix A).

#### 2.3.6. Shoot and Root Na/K Ratio (SNK and RNK)

ICIM and IM detected three and two additive QTLs for SNK under alkaline stress conditions, respectively (Table 3 and Appendix A). The *qSNK9.19* was identified in both ICIM and IM and accounted for 9 and 7% of total PVE, respectively. Other QTLs were with minor effects and explained about 6% of total PVE. N22 was responsible for increasing the trait means at all additive QTLs. No QTLs were detected for SNK under the control condition.

The RNK and SNK QTLs were detected in the same genomic position under alkaline stress in both IM and ICIM (Table 3 and Appendix A). Although the QTL *qRNK4.16* was detected by ICIM, QTLs for both RNK and SNK were detected in the same genomic location in the IM analysis with a contribution of 7% toward PVE. N22 alleles for all the QTLs contributed to the increased trait means. In the control condition, two additive QTLs were identified with desirable alleles contributed by N22 and one QTL was common between IM- and ICIM-detected QTLs.

### 2.4. Co-Localization of QTLs

Comparison of our QTL results with earlier studies revealed that several QTLs were congruent to earlier reported salinity and alkalinity tolerance QTLs (Table 4). The QTL *qSNC3* for shoot Na^+^ concentration, identified under alkaline stress [24], co-localized with *qRKC3.32* detected in this study. Three co-localized QTLs (*qAKT3.03*, *qSNK3.03*, and *qRNK3.03*) in this study were congruent with an earlier-reported salt-responsive root length QTL, *qRTL3.1* [28]. Another QTL cluster on chromosome 8.002 region harboring QTLs for AKT, SNK, RNK, SNC, SKC, RNC, SNC, and CHL (Table 3) was localized to the same chromosomal region as *qNa8.1* for shoot Na^+^ concentration under salt stress [28]. Similarly, the chromosome 9.19 cluster of six QTLs for AKT, SNC, RNC, RKC, SNK, and RNK co-localized with two earlier-reported QTLs, *qNAUP-9a* and *qDWRO-9a*, for Na^+^ uptake and dry root weight under salt stress, respectively [29].

Several QTLs for SHL and RSR colocalized with earlier reported QTLs. The *qSHL1.38* of this study was similar to the *qSHL1.38* [30] and *qPH1.2* [31] under saline stress. Two more QTLs, *qSHL3.13* and *qSHL6.26*, were congruent with shoot dry weight QTL *qDWSH-3* and Na^+^ uptake QTL *qNAUP-6*, respectively, identified under saline stress [29]. The *qSHL6.26* co-localized with *qRSH6*, for relative seedling height under alkaline stress [21]. The *qSHL7.05* was located within the same chromosomal region as *qSDS7*, controlling seedling survival days under saline stress [32]. Another QTL, *qSHL8.27*, was mapped in the identical position as *qRTL8.27* detected in an earlier study [30]. The *qRSR1.35* located at the 35,776,217–37,068,548 bp position on chromosome 1 co-localized with two QTLs, *qRNTQ-1* (root Na^+^ total quantity) and *qSDS1* (survival days for seedling), identified under saline stress [32]. A dead leaf rate QTL, *qDLR3* [22], under alkaline stress had an overlapping QTL region with two QTLs, *qSHL3.13* and *qRSR3.15*, detected in this study.

### 2.5. Gene Ontology Analysis

A total of 1317 candidate genes were identified in 42 QTL intervals for 11 traits under alkaline stress, with an average of 31 genes per QTL (Appendix A). The number of candidate genes within the QTL interval ranged from 1 to 559. There was only one gene for *qRSR1.30*, whereas 559 genes were present in the *qRKC3.32* interval. Among all identified genes, 61% were annotated (Appendix A). There were 334 significant gene ontology terms for all traits (Appendix A). Among 334 GO terms, 191 terms were categorized under biological process and 103 terms for molecular function. The QTL clusters on chromosomes 8, 9, and 10 had 50, 59, 59 significant GO terms, respectively (Table 5). Similarly, 22 and 39 significant GO terms were identified for the QTLs clustered on chromosome 4 (*qSNC4.16*, *qRNK4.16*) and 12 (*qSNC12.19*, *qRNC12.19*), respectively. Two significant GO terms were detected for *qAKT5.008* and *qSKC5.008*, whereas 27 GO terms were identified for *qAKT3.03*, *qSNK3.03*, and *qRNK3.03*.

### 2.6. Discovery of Genotype-Specific SNPs, Indels, and Candidate Genes in the Selected QTL Regions

Three QTL regions were selected for the identification of polymorphic SNPs and indels between Cocodrie and N22 in the respective QTL regions. The chromosome 8.002 QTL cluster had eight co-localized QTLs (*qAKT8.002*, *qRNC8.002*, *qRKC8.002*, *qSKC8.002*, *qRNK8.002*, *qCHL8.002*, *qSNC8.002*, and *qSNK8.002)*, whereas a group of six QTLs (*qAKT9.19*, *qSNC9.19*, *qRNC9.19*, *qRKC9.19*, *qSNK9.19*, and *qRNK9.19)* were located in the chromosome 9.19 region. The *qSKC10.18* region was chosen because it was a major QTL with a high LOD that overlapped with *qAKT10.18*. The SNPs and indels in the genomic regions other than genic regions, such as those 2 kb upstream and downstream or intergenic regions, were filtered out to identify variants. A total of 82 and 613 high- and moderate-impact variants were identified between Cocodrie and N22 for the above three QTL regions (Appendix A). Twenty candidate genes carrying high impact polymorphic SNPs or indels were selected (Table 6). In the chromosome 8.002 region, five frameshift mutations and a stop-gain mutation in two expressed protein genes (LOC_Os08g01560, and LOC_Os08g01720) differentiated Cocodrie and N22 for. Three candidate genes, a glucan endo-1,3-beta-glucosidase precursor (LOC_Os09g32550), *OsFBX387* (LOC_Os0932860), and an expressed protein gene (LOC_Os09g32890) were identified within the chromosome 9.19 region based on the presence of a stop-gain and frameshift mutation, and two splice acceptor variants differentiating Cocodrie and N22. In case of the chromosome 10.18 region, there were fifteen candidate genes which had variants differentiating Cocodrie and N22. These variants carried stop-gain, stop-loss, frameshift, splice acceptor, and splice donor mutations (Table 6).

### 2.7. Expression Pattern of Candidate Genes Located in the Alkalinity Stress Tolerance QTL Intervals Using qRT-PCR

Eight representative genes out of twenty genes from three QTL clusters carrying high-impact variants were selected to determine the expression pattern in response to alkalinity stress by qRT-PCR (Appendix A). Among them, LOC_Os10g35170 (semialdehyde dehydrogenase, NAD binding domain-containing protein), LOC_Os10g35040 (receptor kinase-like protein), LOC_10g33970 (double-stranded RNA-binding motif-containing protein), and LOC_Os08g01720 (expressed protein) were upregulated in Cocodrie compared with N22 under alkaline stress (Figure 5). The expression levels of these genes increased sharply after 6 h of exposure to alkaline stress in Cocodrie, whereas it decreased in N22. There was downregulation of LOC_Os08g01560 (expressed protein), LOC_Os09g32550 (glucan endo-1,3-beta-glucosidase precursor), and LOC_10g34000 (aquaporin protein) in Cocodrie under alkaline stress. However, the expression level of these genes increased in N22. The expression of LOC_Os09g32860 (*OsFBX335*) decreased under stress in both the cultivars.

## 3. Discussion

Soil alkalinity is an important environmental stressor that impairs crop growth and development, resulting in reduced crop productivity. Although alkaline and salinity stress are characteristically different based on ion composition and pH, both are often interconnected and may elicit mixed responses in plants. Compared with salt stress, the impact of alkaline stress is more severe on root growth, nutrient availability, inorganic ionic imbalances, and cellular metabolism [6,33]. However, limited studies have focused on the molecular basis of tolerance to alkalinity stress in rice [22,23,24,25,26,34], which is not only the major staple food for half of the world’s population, but also a valuable model among the Poaceous crops. Therefore, deciphering the genetics of alkaline tolerance in rice is imperative to breed high-yielding rice cultivars with enhanced alkalinity tolerance.

High Na^+^ concentrations are damaging to plant growth, development, and survival. However, plants exhibiting alkalinity or salinity tolerance sequester Na^+^ ions in vacuoles to tolerate high ion concentrations [35,36]. This observation was supported by the significant positive correlation between AKT and Na^+^ concentrations in both shoots and roots in our study. The increased Na^+^ uptake indirectly affected K^+^ transport in plants, which is further supported by the significant negative correlation between Na^+^ and K^+^ concentrations (Table 2). There was a positive correlation between SNC and SNK. A similar trend was observed in previous alkalinity tolerance evaluations involving a set of *japonica* rice germplasms for a GWAS study [25]. The significant negative correlation between AKT and SKC/RKC indicated that accumulation of K^+^ in roots and shoots improved tolerance to ionic stress.

In the case of all SKC and RKC QTLs, Cocodrie alleles were responsible for increasing K^+^ uptake in both shoots and roots. The finding that tolerant parental alleles contributed to increased K^+^ concentrations was consistent with earlier QTL mapping results [11] and implied the desirability of the Cocodrie allele in improving alkalinity tolerance. The presence of transgressive segregants with high K^+^ concentration and low Na^+^ levels and Na^+^/K^+^ ratios signify a contribution of both parental desirable alleles for cation transport under stress. This was evident from the QTL alleles of both parents contributing to enhanced tolerance to alkalinity stress.

In this study, both parents differed with respect to all eleven morpho-physiological traits measured under alkaline stress, but not under the control condition, except for the RNC (Table 1). Alkaline tolerance scoring reflects the overall performance of a line under alkaline stress. The negative correlation between Na^+^ and root and shoot length indicated that increasing sensitivity to alkaline stress results in growth retardation. The QTLs, *qAKT9.19*, *qAKT8.002*, *qAKT10.18*, and *qAKT3.03*, had the AKT-increasing allele from N22, which suggests the importance of the Cocodrie allele at these loci in improving alkalinity tolerance.

Although there was large increase in Na^+^ concentration and decrease in K^+^ concentration under alkaline stress compared with control, Cocodrie showed an increased uptake of K^+^ and reduced uptake of Na^+^ compared with N22, suggesting the superiority of the Cocodrie allele over the N22 allele. This was also reflected in the additive effect of QTLs for the traits in this study (Table 3). Tolerance to alkalinity stress in Cocodrie was most likely due to a lower Na^+^/K^+^ ratio or Na–K homeostasis resulting from the lower accumulation of Na^+^ ions and higher accumulation of K^+^ in roots and shoots [13] and reduced accumulation and effective sequestration of Na^+^ outside of the roots [37]. Another reason for this discrepancy could be the different set of genes or differential expression or induction of genes controlling the transport of these two ions in the parents with a contrasting response to alkalinity stress. High heritability for the uptake of Na^+^ and K^+^ and Na^+^/K^+^ ratio suggested that alkalinity tolerance in rice could be achieved through the selection of these traits using molecular markers.

Tolerance to salinity and alkalinity is a genetically complex trait. Although many genes involved in salt tolerance mechanisms have been identified, very little attention has been given to alkalinity tolerance. As both stresses occur often together, we compared the results from this study with previous QTL mapping studies on salinity and alkalinity tolerance based on the physical map locations (Table 4). Although there were more QTLs co-localized with previously reported salt tolerance QTLs, only few coincided with alkali tolerance QTLs [21,22,24]. The QTL clusters in the 8.002 and 9.19 regions of the rice genome overlapped with QTLs for *qNA8.1* and *qCHL8.1* [28], and *qNAUP-9a* and *qDWRO-9a* [29]. Some novel QTLs were identified in chromosomes 4 (*qRNK4.16* and *qSNC4.16)*, 5 *(**qAKT5.008* and *qSKC5.008)*, 10 *(qAKT10.18* and *qSKC10.18)*, and 12 *(qRNC12.19* and *qSNC12.19).* Though most of these were minor QTLs, *qSKC10.18* was a large-effect QTL, detected with a LOD of 6 and 13% contribution toward phenotypic variation.

There were several regions where QTLs for several traits co-localized. However, we focused on three clusters (9.19, 8.002, and 10.18) to identify candidate genes in the QTL intervals, based on the presence of variants differentiating both parents. Comparison of the variants present in those genes, based on whole genome sequence analysis, narrowed down the candidate genes within the QTL regions. There were two expressed proteins (LOC_Os08g01560 and LOC_Os08g01720) in the 8.002 cluster, whereas three genes, the glucan endo-1,3-beta-glucosidase precursor (LOC_Os09g32550); *OsFBX335*, an F-box-containing protein (LOC_Os09g32860); and an expressed protein (LOC_Os09g32890) were present in the interval of the 9.19 cluster (Table 6). The qRT-PCR analysis revealed that LOC_Os09g32550, LOC_Os09g32860, and LOC_Os08g01560 were downregulated in Cocodrie in comparison with N22 (Figure 5), suggesting their role as negative regulators for alkalinity tolerance in rice. These observations were supported by differential expressions of a glucan endo-1,3-beta-glucosidase gene in the root tissue under saline stress in rice [38] and reduced abiotic stress tolerance in rice due to overexpression of the F-box protein gene [39]. The gene LOC_Os08g01720 was upregulated in Cocodrie, but down regulated in N22. Therefore, the roles of these genes underlying alkalinity tolerance at the seedling stage in rice requires further validation.

In case of the 10.18 genomic region harboring a major-effect *SKC* QTL, there were fifteen potential candidate genes with high-impact DNA polymorphisms between Cocodrie and N22 (Table 6). Among them, two genes, LOC_Os10g34960 and LOC_Os10g34990, were from the ubiquitin protein family, which conferred abiotic stress tolerance in transgenic plants [40]. A gene encoding receptor kinase-like protein (LOC_Os10g35040) was also present within *qSKC10.18.* There were two genes encoding mitochondrial precursor *Rf1*, LOC_Os10g35230 and LOC_Os10g35640, and the latter was downregulated under saline stress [41]. A phosphate translocator gene (LOC_Os10g34490) responsible for phosphate homeostasis in rice was expressed in the root cortex under low phosphorus conditions [42]. Other candidate genes were for the double-stranded RNA-binding motif-containing protein (LOC_Os10g33970), aquaporin protein (LOC_Os10g34000), *OsFBX387* (LOC_Os10g34300), semialdehyde dehydrogenase, and NAD-binding domain-containing protein (LOC_Os10g35170). As these genes have not been implicated in alkalinity tolerance, we selected four genes from the 10.18 QTL region for expression analysis under alkaline stress (Figure 5). Several studies showed that overexpression of aquaporin genes has a negative impact on salt and alkaline stress [43,44]. These findings further corroborate the downregulation of LOC_Os10g34000 in the tolerant cultivar used in this study. Other important candidate genes were for receptor kinase-like protein, putative, expressed (LOC_Os10g35040); semialdehyde dehydrogenase, a NAD-binding domain-containing protein, putative, expressed (LOC_Os10g35170); and double-stranded RNA-binding motif-containing protein, expressed (LOC_Os10g33970), which were all upregulated in Cocodrie compared with N22 under alkaline stress. A previous study showed differential expressions of cysteine-rich receptor-like kinase proteins (CPK) in rice under alkaline stress [34]. Several RNA-binding proteins were associated with tolerance to drought and other abiotic stresses in rice [45,46,47]. Likewise, aldehyde dehydrogenase family genes have been implicated in abiotic stress tolerance in multiple species [48,49,50].

Besides these three genomic regions, several candidate genes associated with the abiotic stress response were also identified. The *qSNC12.19* had candidate genes encoding the integral membrane protein and zinc finger family within its confidence interval, and these proteins were involved in excluding Na^+^ under saline and alkaline stress [51,52]. Abiotic stress tolerance-related genes encoding a calcium-binding protein and putative nucleoporin were present in the *qAKT3.03* region [53,54].

Genes present within the confidence interval of *qRKC3.32*, which was congruent with *qSNC3* [24] and localized adjacent to *qRRL3* and *qADS3* [21], were for a vesicle transport protein (LOC_Os03g57760), cupin domain-containing protein (LOC_Os03g57960), Ca^++^-binding protein (LOC_Os03g59590), membrane-associated DUF588 protein (LOC_Os03g60250), and *ATCHX* (LOC_Os03g61290) (Appendix A). These genes were associated with enhanced abiotic stress tolerance [53,55,56]. A cation hydrogen exchange (CHX) gene improved alkalinity tolerance in soybeans [57]. Other important genes present within the intervals of *qRSR3.15*, *qSHL1.38*, *qSHL3.13*, and *qSHL6.26* were calcium-dependent protein (LOC_Os03g27280), AP2 domain-containing protein (*LOC_Os01g66270*), zinc-finger domain-containing protein (LOC_Os03g24184), and cytokinin-O- glucosyltransferase 3 protein (LOC_Os03g24430), WRKY28 (LOC_Os06g44010), which have been implicated in the abiotic stress response [58,59,60,61,62].

In a GWAS study of a set of *japonica* rice varieties [25], a candidate gene, *OsIRO3* (LOC_Os03g26210), for alkali tolerance was identified in the interval for common QTLs for traits AKT, SNC, and SNK. A 7-bp indel in this gene, which is a negative regulator of the Fe-deficiency response in rice [63], differentiated alkali-tolerant and alkali-susceptible rice varieties. However, this gene was not present in the QTL intervals in this study.

Although alkaline stress is considered similar to salinity stress, several studies have shown that the extent of damage to plants is much greater under alkaline stress than with saline stress. This study demonstrated that rice cultivar Cocodrie is tolerant to alkaline stress, which had previously been shown to be susceptible to saline stress [10]. This observation suggests that tolerance mechanisms for both stresses are different, and a different set of genes confer alkalinity tolerance. The candidate genes identified in this study should be evaluated in the future for their involvement in alkalinity tolerance mechanisms in rice.

## 4. Conclusions

Our study demonstrated the utility of integrating QTL mapping with WGS to identify candidate genes associated with alkalinity tolerance. In addition to novel QTLs, some QTLs identified in this study co-localized with salinity and alkalinity tolerance QTLs from previous studies. There were several clusters of QTLs for the traits used to assess alkalinity tolerance. The QTLs and candidate genes, particularly those originating from the tolerant parent Cocodrie, should be targeted for introgression to improve alkalinity tolerance in rice.

## 5. Materials and Methods

### 5.1. Choice of Parents and Mapping Population

A mapping population consisting of 193 RILs was developed from the cross between Cocodrie and N22. Cocodrie is an agronomically superior US cultivar released by Louisiana State University Agricultural Center [64] and is tolerant to alkaline stress. N22 is a well-known drought tolerance donor [65] but is susceptible to alkaline stress. F_1_ plants from the Cocodrie x N22 cross were selfed to generate F_2_ population, which was then advanced to F_8_ generation by the single seed descent method.

### 5.2. Evaluation of Rice Seedlings for Alkalinity Tolerance

The RIL population, along with parents, were screened at the seedling stage in the LSU AgCenter greenhouse. This experiment was conducted in a randomized complete block design with three replications. There were two sets of experiments: control and stress. In the control experiment, seedlings were continuously grown without any stress, whereas plants in the stress experiment were exposed to alkaline stress at the two-leaf stage. Four-inch pots filled with sand were used for this experiment. Seeds from each line were kept at 50 °C for 4 days to break dormancy. Ten seeds were planted in each pot and a nutrient solution containing 1g/L of Jack’s professional fertilizer (20-20-20) (JR Peters Inc., Allentown, PA, USA) and 300 mg/L ferrous sulphate with 5.6 pH was used. At the two-leaf stage, 0.20 and 0.40% sodium carbonate (Na_2_CO_3_) solution with a pH of 10.0 were used for the first two weeks and third week of stress, respectively. Nutrient solution was replaced every three days, and pH was adjusted to 10.0. Uniform plants were selected for observations on morpho-physiological traits. Chlorophyll content of leaves was measured 10 days after exposure to stress by using the SPAD-502 chlorophyll meter (Spectrum Technologies, Inc., Aurora, IL, USA). The visual alkaline tolerance score was recorded for each line three weeks after exposure to alkaline stress. Alkalinity tolerance scoring was done on a scale of 1–9 based on the percentage of dry and yellow leaves. Alkaline tolerance scores of 1, 2, 3, 4, 5, 6, 7, 8, and 9 were given to seedlings of each line with <20, 21–30, 31–40, 41–50, 51–60, 61–70, 71–80, 81–90, and >90% of dry and yellow leaves, respectively [25]. The data on the shoot length, root length, and root-to-shoot ratio of seedlings were recorded. The root and shoot samples were collected for Na^+^ and K^+^ measurements. The samples were oven-dried at 60 °C for 10 days and 0.5 g of homogenized sample from each line was digested with 5 mL nitric acid and 3 mL hydrogen peroxide at 152–155 °C for 3 h [66]. The amount of Na^+^ and K^+^ in each sample was measured by using a flame photometer (model PFP7, Bibby Scientific Ltd., Staffordshire, UK). Final concentrations of Na^+^ and K^+^ in the shoots and roots were calculated from the standard curve derived from the solutions with different dilutions. Na^+^ to K^+^ ratios in shoots and roots were calculated using Na^+^ and K^+^ concentrations.

### 5.3. Statistical Analysis

The mean values from each replication were used for analysis. Descriptive statistics were obtained using R [67]. The analysis of variance (ANOVA) for each trait under both stress and non-stress environment was computed using the aov function. Pearson correlation coefficients were computed to determine the relationship among different morpho-physiological traits under alkaline stress. Histograms were constructed in R to show the distribution of RILs for each trait. Broad-sense heritability was estimated on a family basis in SAS [68] using the procedure of Holland et al. [69].

### 5.4. Linkage Mapping and QTL Analysis

Previously generated 4748 SNPs by genotyping-by-sequencing in the Cocodrie × N22 RIL population were used for the construction of the linkage map and QTL mapping [70]. As the mapping population was a RIL population, the N22 allele, Cocodrie allele, and any missing data were scored as ‘2’, ‘0’, and ‘−1’, respectively. The GBS genotypic data was used for generation of the linkage map using QTL IciMapping software v. 4.1 [71]. The SNP markers were grouped based on a physical map of the reference genome, Nipponbare. Interval mapping (IM) and inclusive composite interval mapping (ICIM) were performed for QTL analysis using IciMapping software [71]. Mean phenotypic data was used for QTL mapping. A scanning window size of 1 cM with a LOD score of 2.0 was used to declare the significant additive QTLs by ICIM and IM. The phenotypic variation explained by QTLs and their additive effects were estimated. Significant QTLs were named based on the trait name followed by chromosome number and their physical map position on the genome. For example, *qAKT3.03* represents the QTL for alkalinity tolerance score on chromosome 3 at the 3 Mb position. The additive effect of the QTL was used to determine the source parental allele contributing toward the increased trait mean.

### 5.5. Whole Genome Resequencing of Parents

Leaf samples of both parents were collected from 14 d old seedlings and genomic DNA was extracted using a Qiagen DNeasy kit (Qiagen Inc., Valencia, CA, USA). DNA quality and quantity were assessed using a Qubit 2.0 fluorometer (Invitrogen Life Technologies, Eugene, OR, USA) and Bioanalyzer 2100 (Agilent Technologies, Singapore), respectively. The libraries were prepared using the Illumina TruSeq DNA sample preparation kit (Illumina, San Diego, CA, USA) and the paired-end sequencing was performed in the Illumina HiSeq 2000 platform at the Virginia Bioinformatics Institute, Blacksburg, VA, USA. The raw data were filtered using an in-built standard Illumina pipeline.

### 5.6. Read-Mapping and Detection of SNPs and Indels

The FASTQ files from the Illumina pipeline were further analyzed using the NGS QC toolkit v2.3.3 [72] to remove the adapter or primer sequences and low-quality reads. High-quality reads (Phred quality score ≥ 30) were used for mapping. Mapping of high-quality reads was done using Burrows–Wheeler Alignment (BWA v0.7.17), using the mem command with the –M option [73]. SAM files obtained were then sorted and converted to a BAM file using SAM tools v1.12 [74]. Picard tools included in the Genome Analysis Toolkit (GATK v4.0) were used to process the BAM files (FixMateInformation, AddOrReplaceReadGroups, and Mark Duplicates) prior to variant calling [75]. After BAM pre-processing, genomic variants in GVCF format were identified for Cocodrie and N22 using the Haplotype Caller tool in GATK [76]. GVCF files were merged using the CombineGVCFs tool, then performed SNP and INDEL discovery with GenotypeGVCFs. Variant filtering was done using the VariantFiltration tool to apply hard filtering to sites with a quality depth (QD) below 2, strand odds ratio (SOR) above 3, fisher strand (FS) above 60, mapping quality (MQ) below 40, mapping quality tank sum (MQRankSum) below −12.5, and read position rank sum (ReadPosRankSum) below −8 for SNPs, and a QD below 2.0, FS above 200, and ReadPosRankSum below −20 for INDELs, as described in the GATK best practices recommendation on hard filtering. Annotation of the variants was done using SnpEff v5.0e [77]. Variant sites were restricted to polymorphic alleles between Cocodrie and N22, and excluded sites with synonymous, upstream, downstream, intron, and intergenic variant effects. Candidate genes were then selected from the three selected QTL regions.

### 5.7. Gene Ontology and Annotation

QTL intervals were determined based on the physical position of left and right flanking markers. The candidate genes present in QTL confidence intervals were retrieved from the MSU rice genome annotation project database (http://rice.plantbiology.msu.edu/ Accessed on 1 June 2021) for all the QTLs identified in this study. Genes present in the QTL intervals were identified using the position of flanking SNP markers on the genome. All candidate genes present within the QTL regions were annotated using the AgriGO toolkit to determine the functional relevance of the genes in alkalinity tolerance mechanisms in rice [78]. Annotation of the identified SNPs and indels for the three QTL regions and prediction of variant effects (low, moderate, and high) were done using SnpEff (v.4.2) software [77].

### 5.8. Validation of Selected Genes in the QTL Interval by Quantitative Reverse Transcription PCR (qRT-PCR)

Cocodrie and N22 seeds were grown in a hydroponic experiment containing 1 g/L of Jack’s professional fertilizer (20-20-20) (JR Peters Inc., Allentown, PA, USA). The two sets of experiments (control and stress) were conducted with three replications. Twenty seedlings were grown in each replication. At the two-leaf stage, the seedlings of Cocodrie and N22 were subjected to alkaline stress treatment using 0.5% Na_2_CO_3_ solution with pH 10.0. Leaf samples were collected at 0 and 6 h after imposition of stress from both sets of experiment. Collected leaf tissues were placed in liquid N during collection and stored in a −80 °C freezer until RNA extraction. The total RNA from leaf tissues of three biological replicates per treatment was isolated using Trizol reagent (Thermofisher Scientific, Waltham, MA, USA). Quality of the total RNA was checked on 1.2% agarose gel and quantity of the RNA was evaluated using a ND-1000 spectrophotometer (Thermofisher Scientific, Waltham, MA, USA). The samples were then treated with PerfeCTa DNase 1 (Quantabio, Beverly, MA, USA). First strand cDNA was synthesized using iScript™ first strand cDNA synthesis kit (Bio-Rad Laboratories, Hercules, CA, USA), as per the manufacturer’s instructions. The qRT-PCR reaction was used to evaluate the expression of genes present in the selected QTLs that carried high-impact variants based on the whole genome data of both parents. The primers were designed using the PrimerQuest Tool (Integrated DNA Technologies, Inc., Coralville, IA, USA) (Appendix A). The elongation factor 1 alpha (*EF1α*, LOC_Os03g08010) was used as an internal control. The qRT-PCR reaction was performed in three technical replicates using cDNA pooled from the biological replicates following the previously described protocol [79]. A total reaction of 10 µL was set up on a QuantStudio 3 Real-Time PCR system (Applied Biosystems Corporation, Waltham, MA, USA) using iTaq™ Universal SYBR Green Supermix (Bio-Rad Laboratories, Hercules, CA, USA). The expression level of genes was determined using the 2^–∆∆CT^ method [80]. After normalizing the Ct values using the internal control gene *EF1α*, we calculated the fold change in expression of genes under alkalinity stress compared with control for both parents.

## Figures and Tables

**Figure 1 ijms-23-11791-f001:**
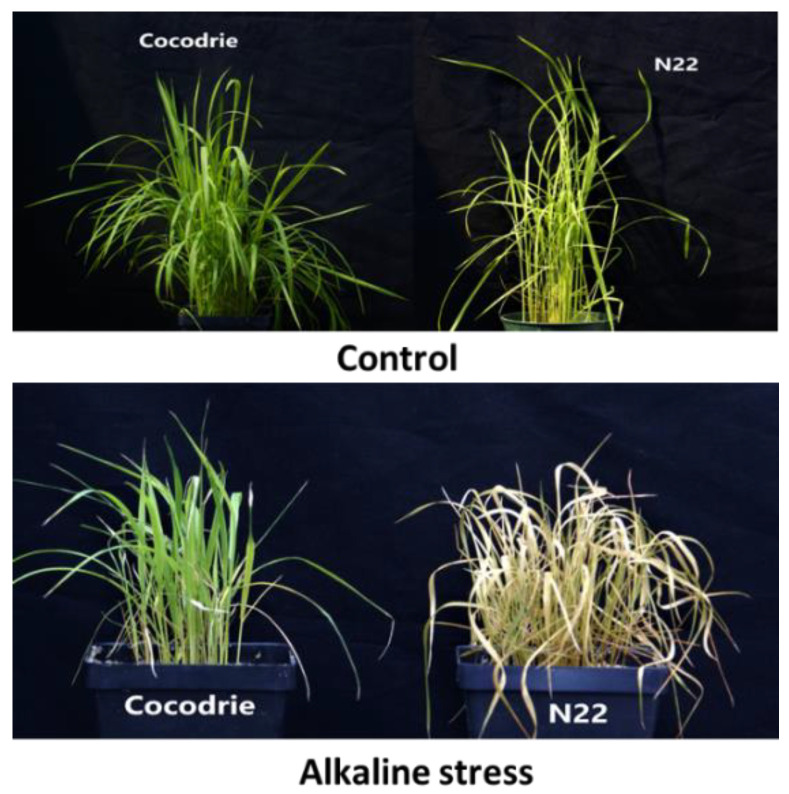
Comparison of performance of Cocodrie and N22 under (upper panel) control and (lower panel) alkaline stress conditions, respectively.

**Figure 2 ijms-23-11791-f002:**
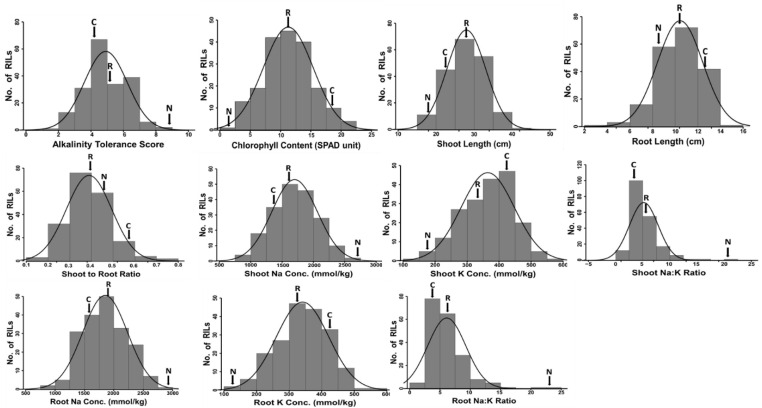
Frequency distribution of Cocodrie/N22 RIL population for seedling stage alkalinity tolerance for various morphological and physiological traits. The arrowhead indicates the trait mean for Cocodrie (C), N22 (N), and RIL population (R).

**Figure 3 ijms-23-11791-f003:**
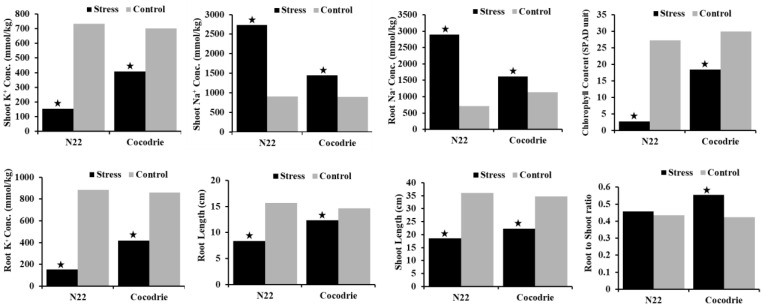
Comparison of Cocodrie and N22 for various physiological and morphological traits under alkaline stress and non-stress environments. Single asterisk indicates significant difference between the means of Cocodrie and N22 for stressed and control environment at 0.05 level of probability.

**Figure 4 ijms-23-11791-f004:**
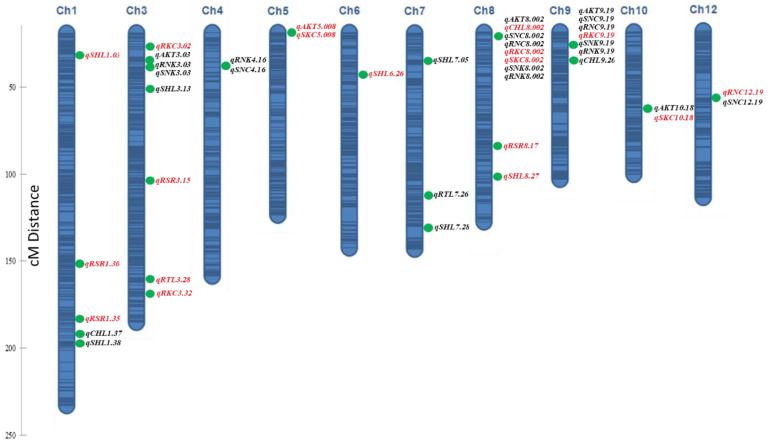
Map positions of the QTLs for eleven morphological and physiological traits in the Cocodrie × N22 RIL population. N22 and Cocodrie alleles responsible for the increased mean are indicated in black and red font, respectively. Dark regions on the genetic map are the marker-saturated regions; light regions represent gaps between the markers.

**Figure 5 ijms-23-11791-f005:**
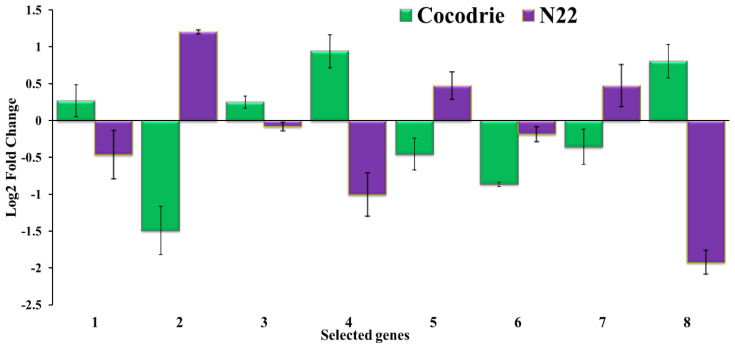
Expression profiles of eight selected genes present in the alkalinity tolerance QTL regions under alkalinity stress (6 h after imposition of stress) in Cocodrie and N22. The selection was based on differences in high-impact variants between the parents. Selected genes included: 1-LOC_Os10g35040 (receptor kinase-like protein); 2-LOC_Os10g34000 (aquaporin protein); 3-LOC_Os10g35170 (semialdehyde dehydrogenase, NAD-binding domain-containing protein); 4-LOC_Os10g33970 (double-stranded RNA-binding motif-containing protein); 5-LOC_Os09g32550 (glucan endo-1,3-beta-glucosidase precursor); 6-LOC_Os09g32860 (*OsFBX335*-F-box domain-containing protein); 7-LOC_Os08g01560 (expressed protein); and 8-LOC_Os08g01720 (expressed protein). *EF1α* was used as the reference gene and gene expressions were expressed as log2 fold changes under alkaline stress compared with control in both parents.

**Table 1 ijms-23-11791-t001:** Mean phenotypic performance of the Cocodrie x N22 RIL population for various morphological and physiological traits at the seedling stage, under non-stress or alkaline stress environments.

Alkaline Stress						Control				Reduction in Trait Mean (%) Under Stress ^d^	
Trait ^a^	Cocodrie Mean	N22 Mean ^b^	RIL Mean ^c^	RIL Range	h^2^	Cocodrie Mean	N22 Mean ^b^	RIL Mean	RIL Range	Cocodrie	N22
AKT	4.3	9.0 **	5.0 **	2.0–9.0	0.7	1.0	1.0 ^ns^	1.1	1.0–3.0	−76.7	−88.9
CHL (SPAD)	18.4	2.7 *	11.3 *	2.2–22.1	0.5	29.9	27.2 ^ns^	25.5	18.2–34.2	38.5	90.1
SHL (cm)	22.2	18.5 *	28.1 **	17.2–40.1	0.8	34.7	36.0 ^ns^	37.8	20.3–55.3	35.9	48.6
RTL (cm)	12.3	8.3 *	10.4 **	3.9–14.1	0.7	14.7	15.7 ^ns^	12.7	5.5–16.5	16.1	47.0
RSR	0.6	0.4 **	0.4 **	0.2–0.8	0.7	0.4	0.5 ^ns^	0.4	0.2–0.6	−35.7	2.2
SNC (mmol/Kg)	1448.2	2734.2 **	1690.4 **	832.4–2734.2	0.9	866.7	900.0 ^ns^	944.0	508.3–1383.3	−67.1	−203.8
SKC (mmol/Kg)	408.6	153.6 **	365.1 **	149.2–592.3	0.8	711.3	720.0 ^ns^	755.2	406.7–1106.7	42.6	78.7
RNC (mmol/Kg)	1619.2	2900.1 **	1851.7 **	903.3–2900.1	0.8	1106.0	657.1 **	1048.4	564.5–1536.3	−46.4	−341.3
RKC (mmol/Kg)	431.0	131.3 **	342.6 **	131.3–556.2	0.9	779.3	809.3 ^ns^	848.8	124.3–457.1	44.7	83.8
SNK	3.5	20.8 **	5.1 **	1.4–20.8	0.9	1.2	1.3 ^ns^	1.5	1.1–2.9	−192.6	−1565.6
RNK	3.8	23.0 **	6.0 **	1.6–23.0	0.9	1.4	0.8 ^ns^	1.4	0.6–3.1	−166.0	−2739.5

^a^ AKT, alkalinity tolerance scoring; CHL, chlorophyll content; SHL, shoot length; RTL, root length; RSR, root-to-shoot ratio; SNC, shoot Na^+^ concentration; SKC, shoot K^+^ concentration; RNC, root Na^+^ concentration; RKC, root K^+^ concentration; SNK, shoot Na^+^: K^+^ ratio; RNK, root Na^+^: K^+^ ratio. ^b^
*t*-test between Cocodrie and N22; ^c^ genotypic difference among RILs; ^d^ negative and positive values indicate increase and decrease in trait mean, respectively. *, ** significant differences between the means for Cocodrie and N22 at 0.05 and 0.01 level of probability, respectively. ^ns^ nonsignificant. h^2^—heritability.

**Table 2 ijms-23-11791-t002:** Pearson correlation matrix of phenotypic traits measured in the Cocodrie × N22 RIL population at the seedling stage under alkaline stress.

Trait ^a^	AKT	CHL	SHL	RTL	RSR	SNC	SKC	RNC	RKC	SNK	RNK
AKT	1.00										
CHL	−0.77 **	1.00									
SHL	−0.02	−0.11	1.00								
RTL	−0.16 *	0.11	0.05	1.00							
RSR	−0.09	0.17 *	−0.64 **	0.69 **	1.00						
SNC	0.98 **	−0.76 **	−0.01	−0.15 *	−0.09	1.00					
SKC	−0.98 **	0.76 **	0.02 **	0.18 **	0.11	−0.96 **	1.00				
RNC	0.97 **	−0.75 **	−0.01 *	−0.14 *	−0.09	0.99 **	−0.96 **	1.00			
RKC	−0.96 **	0.74 **	−0.00 **	0.19 **	0.14 *	−0.94 **	0.98 **	−0.94 **	1.00		
SNK	0.92 **	−0.68 **	−0.06	−0.21 **	−0.10	0.91 **	−0.92 **	0.90 **	−0.91 **	1.00	
RNK	0.92 **	−0.68 **	−0.05	−0.20 **	−0.11	0.91 **	−0.92 **	0.90 **	−0.92 **	0.99 **	1.00

^a^ AKT, alkalinity tolerance scoring; CHL, chlorophyll content; SHL, shoot length; RTL, root length; RSR, root-to-shoot ratio; SNC, shoot Na^+^ concentration; SKC, shoot K^+^ concentration; RNC, root Na^+^ concentration; RKC, root K^+^ concentration; SNK, shoot Na^+^: K^+^ ratio; RNK, root Na^+^:K^+^ ratio. * Significant at 0.05 level of probability; ** Significant at 0.01 level of probability.

**Table 3 ijms-23-11791-t003:** List of additive QTLs for various morphological and physiological traits associated with alkaline stress at seedling stage of rice by ICIM mapping.

Trait ^a^	QTL	Chr	Position(cM)	Left Marker	Right Marker	Interval (bp)	LOD ^b^	PVE (%) ^c^	Additive Effect	No. of Genes in QTL Interval	Parental Allele with Increasing Effect
AKT	*qAKT3.03*	3	167	S3_3798053	S3_3978854	180,801	2.40	4.5	0.28	26	N22
	*qAKT5.008*	5	5	S5_865267	S5_891285	26,018	2.30	4.0	−0.26	4	Cocodrie
	*qAKT8.002*	8	125	S8_261276	S8_498009	236,733	3.38	6.4	0.33	37	N22
	*qAKT9.19* *qAKT10.18*	9 10	22 70	S9_19333995 S10_18053155	S9_19696641 S10_19335416	362,646 1,282,261	5.89 2.02	11.6 4.2	0.45 0.26	59 187	N22 N22
CHL	*qCHL1.37*	1	186	S1_37740707	S1_37777636	36,929	3.04	6.3	0.97	4	N22
	*qCHL8.002*	8	125	S8_261276	S8_498009	236,733	2.64	5.3	−0.88	37	Cocodrie
	*qCHL9.20*	9	17	S9_20470185	S9_20519258	49,073	8.27	17.8	1.63	3	N22
SHL	*qSHL1.03*	1	20	S1_3695146	S1_3708821	13,675	2.31	1.8	−0.67	2	Cocodrie
	*qSHL1.38*	1	190	S1_38286772	S1_38611845	325,073	33.95	43.8	3.37	44	N22
	*qSHL3.13*	3	110	S3_13712517	S3_13934642	222,125	3.25	2.7	0.84	20	N22
	*qSHL6.26*	6	29	S6_26499660	S6_26675063	175,403	2.58	2.1	−0.74	19	Cocodrie
	*qSHL7.05*	7	33	S7_5312649	S7_5549169	236,520	2.52	2.0	0.73	28	N22
	*qSHL7.28*	7	139	S7_28852810	S7_28875695	22,885	5.79	5.0	1.13	3	N22
	*qSHL8.27*	8	2	S8_27384352	S8_27875737	491,385	4.00	3.5	−0.95	79	Cocodrie
RTL	*qRTL3.28*	3	52	S3_28513305	S3_28809504	296,199	2.66	6.8	−0.46	41	Cocodrie
	*qRTL7.26*	7	11	S7_26075952	S7_26090857	14,905	3.03	7.7	0.50	1	N22
RSR	*qRSR1.30*	1	147	S1_30155765	S1_30162203	6,438	2.79	4.0	−0.02	1	Cocodrie
	*qRSR1.35*	1	181	S1_35776217	S1_37068548	1,292,331	13.43	21.2	−0.04	94	Cocodrie
	*qRSR3.15*	3	101	S3_15513823	S3_15747509	233,686	4.39	6.1	−0.02	25	Cocodrie
	*qRSR8.17*	8	67	S8_17338253	S8_17443562	105,309	2.47	3.4	−0.01	9	Cocodrie
SNC	*qSNC4.16*	4	40	S4_16612171	S4_16880788	268,617	2.12	4.4	73.20	26	N22
	*qSNC8.002*	8	125	S8_261276	S8_498009	236,733	3.41	8.2	94.03	37	N22
	*qSNC9.19* *qSNC12.19*	9 12	22 68	S9_19333995 S12_19968349	S9_19696641 S12_20375777	362,646 407,428	4.80 2.04	11.8 4.2	113.8 71.79	59 33	N22 N22
RNC	*qRNC8.002*	8	125	S8_261276	S8_498009	236,733	3.60	8.5	103.34	37	N22
	*qRNC9.19*	9	22	S9_19333995	S9_19696641	362,646	4.87	11.6	122.41	59	N22
	*qRNC12.19*	12	68	S12_19968349	S12_20375777	407,428	2.07	4.5	−77.19	33	Cocodrie
SKC	*qSKC5.008*	5	5	S5_865267	S5_891285	26,018	2.07	3.9	−15.81	4	Cocodrie
	*qSKC8.002*	8	125	S8_261276	S8_498009	236,733	4.10	8.4	−23.04	37	Cocodrie
	*qSKC10.18*	10	70	S10_18053155	S10_19335416	1,282,261	6.01	12.6	−28.49	187	Cocodrie
RKC	*qRKC3.02* *qRKC3.32*	3 3	13 106	S3_2264990 S3_32785101	S3_2295597 S3_36366411	30,607 3,581,307	5.33 6.57	6.5 8.3	−19.03 −27.12	5 559	Cocodrie Cocodrie
	*qRKC8.002* *qRKC9.19*	8 9	125 22	S8_261276 S9_19333995	S8_498009 S9_19696641	236,733 362,646	5.66 7.66	7.0 9.8	−24.51 −29.33	37 59	Cocodrie Cocodrie
SNK	*qSNK3.03*	3	21	S3_3978853	S3_4050070	71,217	2.85	6.3	0.57	8	N22
	*qSNK8.002*	8	125	S8_261276	S8_498009	236,733	2.55	5.6	0.54	37	N22
	*qSNK9.19*	9	22	S9_19333995	S9_19696641	362,646	3.77	8.5	0.68	59	N22
RNK	*qRNK3.03*	3	167	S3_3798053	S3_3978854	180,801	2.87	6.3	0.69	26	N22
	*qRNK4.16*	4	40	S4_16612171	S4_16880788	268,617	3.16	6.7	0.79	26	N22
	*qRNK8.002*	8	125	S8_261276	S8_498009	236,733	2.69	5.9	0.67	37	N22
	*qRNK9.19*	9	22	S9_19333995	S9_19696641	362,646	3.87	8.6	0.82	59	N22

^a^ AKT, alkalinity tolerance scoring; CHL, chlorophyll content; SHL, shoot length; RTL, root length; RSR, root-to-shoot ratio; SNC, shoot Na^+^ concentration; SKC, shoot K^+^ concentration.; RNC, root Na^+^ concentration; RKC, root K^+^ concentration; SNK, shoot Na^+^: K^+^ ratio; RNK, root Na^+^:K^+^ ratio. ^b^ LOD logarithm of odd; ^c^ PVE (%) percentage phenotypic variance explained by the QTL.

**Table 4 ijms-23-11791-t004:** List of previously reported QTLs colocalized with the QTLs detected in this study.

	Current Study			
QTL ^a^	Physical Position	QTL	Position (Flanking Markers)	References
*qSHL1.38*	38,286,772–38,611,845	*qSHL1.38* *qPH1.2*	38286772–38611845 164.4–170.3 cM (id1024972–id1025983)	[30,31]
*qSHL3.13*	13,712,517–13,934,642	*qDWSH-3*	7,232,837–16,968,975 (RM1022–RM6283)	[29]
*qSHL6.26*	26,499,660–26,675,063	*qNAUP-6* *qRSH6*	22,297,146–28,599.319 (RM3287–RM340) 26,554,756–28,532,453 (RM454-RM528)	[21,29]
*qSHL7.05*	5,312,649–5,549,169	*qSDS7*	4,573,316–7,739,951 (R2401–L538T7)	[32]
*qSHL8.27*	27,384,352–27,875,737	*qRTL8.27*	27,238,050–27,304,101	[30]
*qRSR1.35*	35,776,217–37,068,548	*qRNTQ-1*, *qSDS1*	33,956,950–37,713,775 (C813–C86)	[32]
*qSHL3.13* *qRSR3.15*	13,712,517–13,934,642 15,513,823–15,747,509	*qDLR3*	13,221,482–20,244,184 (RM338–RM2453)	[22]
*qRKC3.32*	32,785,101–36,366,411	*qSNC3*	33,386,334–35,669,797 (RM1221–RM130	[24]
*qAKT3.03*, *qRNK3.03*, *qSNK3.03*	3,798,053–3,798,854 3,978,853–4,050,070	*qRTL3.1*	3,803,115–5,337,745 (RM5474–RM5480)	[28]
*qAKT8.002*, *qSNK8.002*, *qRNK8.002*, *qSNC8.002*, *qRNC8.002*, *qSKC8.002*, *qRKC8.002 qCHL8.002*,	261,176–498,009	*qNA8.1*, *qCHL8.1*	125,275–4,772,897 (RM408–RM1111)	[28]
*qSNC9.19*, *qRNC9.19*, *qRKC9.19*, *qSNK9.19*, *qRNK9.19*, *qAKT9.19*	19,333,995–19,696,641	*qNAUP-9a* *qDWRO-9a*	16,580,865–21,003,387 (RM1553–RM7424)	[29]

**^a^** AKT, alkalinity tolerance scoring; CHL, chlorophyll content; SHL, shoot length; RSR, root-to-shoot ratio; SNC, shoot Na^+^ concentration; RNC, root Na^+^ concentration; SKC, shoot K^+^ concentration; RKC, root K^+^ concentration, SNK, shoot Na^+^: K^+^ ratio; RNK, root Na^+^:K^+^ ratio.

**Table 5 ijms-23-11791-t005:** Gene ontology analysis of clustered QTL for alkaline stress responsive traits.

QTL Clusters ^a^	Total No. of Genes	No. of Genes Annotated	Annotated Genes (%)	Numbers of Significant Ontology Terms		
				Biological Process	Cellular Components	Molecular Function
*qAKT3.03*, *qRNK3.03*, *qSNK3.03*	26	8	31	18	6	3
*qAKT5.008*, *qSKC5.008*	4	2	50	2	0	0
*qAKT8.002*, *qCHL8.002*, *qSNC8.002*, *qRNC8.002*, *qSKC8.002*, *qRKC8.002*, *qSNK8.002*, *qRNK8.002*	37	26	70	31	13	6
*qAKT9.19*, *qSNC9.19*, *qRNC9.19*, *qRKC9.19*, *qSNK9.19*, *qRNK9.19*	59	37	63	35	16	8
*qAKT10.18*, *qSKC10.18*	187	109	58	48	9	2
*qSNC12.19*, *qRNC12.19*	33	16	48	19	14	6
*qSNC4.16*, *qRNK4.16*	26	14	54	13	5	4

**^a^** AKT, alkalinity tolerance scoring; CHL, chlorophyll content; SHL, shoot length; RTL, root length; RSR, root-to-shoot ratio; SNC, shoot Na^+^ concentration; SKC, shoot K^+^ concentration; RNC, root Na^+^ concentration; RKC, root K^+^ concentration; SNK, shoot Na^+^: K^+^ ratio; RNK, root Na^+^:K^+^ ratio.

**Table 6 ijms-23-11791-t006:** Polymorphic SNPs and indels (high-impact) in the genomic region of three selected QTLs between Cocodrie and N22.

QTL Cluster ^$^	MSU Locus ID	Physical Position ^#^	N22 Allele	Cocodrie Allele	SNP/Indel Annotation ^¥^	Molecular Function
*qAKT8.002* *qCHL8.002* *qSNC8.002* *qRNC8.002* *qRKC8.002* *qSKC8.002* *qSNK8.002* *qRNK8.002*	LOC_Os08g01560	332092	G	^a^ 17-bp	FS	Expressed protein
		332260	A	^b^ 11-bp	FS	
		332300	T	^c^ 14-bp	FS	
		332354	A	^d^ 39-bp	FS	
	LOC_Os08g01720	439569	A	G	SG	Expressed protein
		439759	G	GC	FS	
*qAKT9.19* *qSNC9.19* *qRNC9.19* *qRKC9.19* *qSNK9.19* *qRNK9.19*	LOC_Os09g32550	19437532	C	T	SA	Glucan endo-1,3-beta-glucosidase precursor, putative, expressed
	LOC_Os09g32860	19591049	T	A	SA	OsFBX335—F-box domain-containing protein, expressed
		19592499	CA	C	FS	
	LOC_Os09g32890	19609436	A	C	SG	Expressed protein
*qAKT10.18* *qSKC10.18*	LOC_Os10g33970	18106744	C	G	SG	Double-stranded RNA-binding motif-containing protein, expressed
	LOC_Os10g34000	18141437	C	G	SA	Aquaporin protein, putative, expressed
	LOC_Os10g34300	18295111	A	G	SG	OsFBX387—F-box domain-containing protein, expressed
		18295616	C	CGGCG	FS	
		18296954	T	^e^ 17-bp	FS	
	LOC_Os10g34490	18402218	G	A	SL	Phosphate translocator-related, putative, expressed
		18402757	C	CG	FS	
		18403126	GCGCTCAC	G	FS	
	LOC_Os10g34614	18452230	C	CAA	FS	csAtPR5, putative, expressed
		18453490	A	C	SG	
	LOC_Os10g34960	18658668	C	^f^ 23-bp	FS	Ubiquitin family protein, putative, expressed
		18659948	TC	T	FS	
	LOC_Os10g34990	18666630	T	TCTTC	FS	Ubiquitin-carboxyl extension, putative, expressed
		18666647	ATGCT	A	FS	
		18666931	A	G	SG	
	LOC_Os10g35040	18697585	C	^g^ 10-bp	FS	Receptor kinase-like protein, putative, expressed
	LOC_Os10g35160	18769659	CTT	C	FS	Expressed protein
	LOC_Os10g35170	18774108	A	G	SD	Semialdehyde dehydrogenase, NAD-binding domain-containing protein, putative, expressed
	LOC_Os10g35230	18823689	C	A	SL	Rf1, mitochondrial precursor, putative, expressed
	LOC_Os10g35330	18885478	A	G	SG	Expressed protein
	LOC_Os10g35640	19057187	TCC	T	FS	Rf1, mitochondrial precursor, putative, expressed
	LOC_Os10g35940	19206693	A	AGC	FS	Folylpolyglutamate synthetase, putative, expressed
		19206697	T	TGGTG	FS	
	LOC_Os10g36050	19263325	GAC	G	FS	Hypothetical protein

^$^*qAKT*, *qCHL*, *qSNC*, *qRNC*, *qSKC*, *qRKC*, *qSNK*, and *qRNK* are QTLs for alkalinity tolerance score, chlorophyll content, shoot Na^+^ content, root Na^+^ content, shoot K^+^ content, root K^+^ content, shoot Na^+^: K^+^ ratio, and root Na^+^:K^+^ ratio, respectively. ^#^ Physical position based on IRGSP 1.0. ^¥^ FS, frame shift; SG, stop-gain; SL, stop-loss; SA, splice acceptor; SD, splice donor. ^a^ 17-bp (GCGATGAACCCCCTACT); ^b^ 11-bp (AGATGGTCTCC); ^c^ 14-bp (GCACCGCGCAGTAT); ^d^ 39-bp (TGCTCACCGTCTCAACACACTGAGGGCAACCAAATCCAA); ^e^ 17-bp (TGCTTGTCGGGGAGATC); ^f^ 23-bp (CCCTTCTCCCCGCCGGTCACCAT); ^g^ 10-bp (CGGCGGCGAT).

## Data Availability

The data presented in this study are available in the article and Appendix A.

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
