# Peer review of "Integration of QTL Mapping and Whole Genome Sequencing Identifies Candidate Genes for Alkalinity Tolerance in Rice (Oryza sativa)"

_ijms, 2022, doi:10.3390/ijms231911791_

Round 1

Reviewer 1 Report

Dear authors

I read the article. It is an interesting subject.

General comments

Please answer my questions:

Why the authors used themself references?

What is the objective of this study?

What is the novelty of this study? ( it is not clear to me)

Specific comments

Line 57 despite the fact that change to even though

Line 146 is not clear

Line 222 needs to revision

Line 388 needs to revision

The conclusion needs to revision

Author Response

Reviewer 1

General comments

Please answer my questions:

Why the authors used themself references?

Authors’ response: Our group has been working on salinity tolerance for last ten years. Since the populations used in both salinity and alkalinity studies were developed in the US cultivars background, it was natural for us to compare our results from salinity mapping studies with the alkalinity studies. Another reason is that salinity and alkalinity stresses often occur together, the comparison in the similar genetic background will be more meaningful to ascertain the differences in the genetic basic of tolerance to both salinity and alkalinity stresses.

What is the objective of this study?

Authors’ response: The main objective of this study is to identify alkalinity tolerance QTLs using a high-resolution genetic map followed by integration of the QTL information with whole genome sequence data to identify candidate genes associated with alkalinity tolerance in rice.

What is the novelty of this study? ( it is not clear to me)

Authors’ response: In this study, we identified some novel QTLs for alkalinity tolerance. Although some common genes/QTLs may be involved in alkalinity tolerance as reflected from the colocalization of alkalinity tolerance QTLs with salinity tolerance QTLs, our study showed that genetic basis of alkalinity tolerance is different from salinity tolerance. This is supported by the observation that the alkalinity tolerant parent Cocodrie if highly susceptible to salinity (De Leon et al. 2015). We also demonstrated that combined analysis of whole genome sequencing with QTL information is an effective method to identify potential candidate genes.

Specific comments

Line 57 despite the fact that change to even though

Authors’ response: revised as suggested by the reviewer.

Line 146 is not clear

Authors’ response: We have revised the sentences to make it clear.

Line 222 needs to revision

Authors’ response: Sorry for our oversight. We revised the sentence to indicate that 5 additive QTLs each were identified by ICIM and IM under nonstress condition.

Line 388 needs to revision

Authors’ response: Revised the sentence to increase clarity..

The conclusion needs to revision

Authors’ response: The conclusion is revised.

Reviewer 2 Report

The manuscript entitled “Integration of QTL Mapping and Whole Genome Sequencing Identifies Candidate Genes for Alkalinity Tolerance in rice” is very interesting and authors have selected an important aspect. The population size is ok and data were well analyzed. The authors identified QTLs for alkalinity tolerance in rice seedlings. I have some queries as bellow.

1. Add the scientific name in the title.

2. I have marked the corrections in the manuscript. Please incorporate.

3. Arrange the keywords in alphabetical order.

4. Line-33: Check the font size of number.

5. Line 96-99: Please justify the para.

6. Line 104-105: The sentence is not clear.

7. Please correct the symbol of heritability.

8. The significance test at 0.05 and 0.01 are ok. In my opinion no need to test at 0.001.

9. In table-3: The QTLs had more than 3.0 LOD should be listed here. There is no meaning of lower LOD.

10. I tables, please keep decimal point uniform.

11. The name of QTLs and genes should be itaics. Please check through out manuscript.

12. Line 184: The subsection point missing.

13. Line 328: N2 will be replaced with N22.

14. Line 413: Please check.

15. Line 508-509: Please correct as highlighted in the manuscript.

16. Line 519: Please correct as highlighted in the manuscript.

17. Line 654-820: All references had duplicate sr. Number. Please correct it.

Author Response

Reviewer 2

The manuscript entitled “Integration of QTL Mapping and Whole Genome Sequencing Identifies Candidate Genes for Alkalinity Tolerance in rice” is very interesting and authors have selected an important aspect. The population size is ok and data were well analyzed. The authors identified QTLs for alkalinity tolerance in rice seedlings. I have some queries as bellow.

  1. Add the scientific name in the title.

Authors’ response: Scientific name added in the title.

  1. I have marked the corrections in the manuscript. Please incorporate.

Authors’ response: Thanks for your editing. We have revised the manuscript accordingly.

  1. Arrange the keywords in alphabetical order.

Authors’ response: keywords arranged in alphabetical order

  1. Line-33: Check the font size of number.

Authors’ response: Corrected as suggested by the reviewer.

  1. Line 96-99: Please justify the para.

Authors’ response: This was formatting error by the editorial office. We have corrected as suggested by the reviewer.

  1. Line 104-105: The sentence is not clear.

Authors’ response: Revised to add clarity.

  1. Please correct the symbol of heritability.

Authors’ response: Corrected as suggested by the reviewer.

  1. The significance test at 0.05 and 0.01 are ok. In my opinion no need to test at 0.001.

Authors’ response: We agree with the reviewer. We therefore revised to show significance test at 0.05 and 0.01.

  1. In table-3: The QTLs had more than 3.0 LOD should be listed here. There is no meaning of lower LOD.

Authors’ response: Although we agree with the reviewer, there is no consensus regarding the use of LOD threshold to declare presence of QTLs. There is wide range of variation in literature regarding the LOD threshold to detect QTLs in a variety of plant and crop species. In this study, we used LOD 2.0 as cutoff point to declare the significant QTLs because we did not want to miss potential QTLs associated with alkalinity tolerance. The threshold of LOD 2.0 has been used in several studies (Ming-zhe et al. 2005; Lee et al. 2006; Lee et al. 2007; Amar et al. 2009; Ren et al. 2010; De León et al. 2011; Shang et al. 2016; Asif et al. 2022; Guo et al. 2022) including our own studies.

References:

Ming-zhe et al. (2005) Rice Science 12:25-32.

Lee et al. (2006) Mol Cells 21:192-196.

Lee et al. (2007) Plant Breeding 126:43-46.

Amar et al. (2009) J Plant Biochem Biochem 18:139-150.

Ren et al. (2010) Proc Natl Acad Sci USA 107:5669-5674.

De León et al. (2011) Euphytica 181:371–383.

Shang et al. (2016) G3 Genes|Genomes|Genetics, 6:2717–2724,

Asif et al. (2022) Plants 11(19):2467.

Guo et al. (2022) G3 Genes|Genomes|Genetics 12(6), https://doi.org/10.1093/g3journal/jkac099

  1. I tables, please keep decimal point uniform.

Authors’ response: We revised the table to have uniform decimal points in al tables.

  1. The name of QTLs and genes should be itaics. Please check through out manuscript.

Authors’ response: Thanks for the suggestion. We checked throughout the manuscript and italicized the genes and QTLs.

  1. Line 184: The subsection point missing.

Authors’ response: Subsection points were added and other subpoints revised accordingly.

  1. Line 328: N2 will be replaced with N22.

Authors’ response: N2 was replaced with N22.

  1. Line 413: Please check.

Authors’ response: revised the sentence as suggested by the reviewer.

  1. Line 508-509: Please correct as highlighted in the manuscript.

Authors’ response: Corrected as suggested by the reviewer.

  1. Line 519: Please correct as highlighted in the manuscript.

Authors’ response: Corrected as highlighted by the reviewer.

  1. Line 654-820: All references had duplicate sr. Number. Please correct it.

Authors’ response: Corrected as suggested by the reviewer.